# A Gaussian Process Model of Quasar Spectral Energy Distributions

**Andrew Miller**[*] **, Albert Wu**
School of Engineering and Applied Sciences
Harvard University
acm@seas.harvard.edu, awu@college.harvard.edu

**Jeffrey Regier,   Jon McAuliffe**
Department of Statistics
University of California, Berkeley
{jeff, jon}@stat.berkeley.edu

**Dustin Lang**
McWilliams Center for Cosmology
Carnegie Mellon University
dstn@cmu.edu

**Prabhat,   David Schlegel**
Lawrence Berkeley National Laboratory
{prabhat, djschlegel}@lbl.gov

**Ryan Adams** [†]
School of Engineering and Applied Sciences
Harvard University
rpa@seas.harvard.edu

## Abstract

We propose a method for combining two sources of astronomical data, spectroscopy and photometry, that carry information about sources of light (e.g., stars, galaxies, and quasars) at extremely different spectral resolutions. Our model treats the spectral energy distribution (SED) of the radiation from a source as a latent variable that jointly explains both photometric and spectroscopic observations. We place a flexible, nonparametric prior over the SED of a light source that admits a physically interpretable decomposition, and allows us to tractably perform inference. We use our model to predict the distribution of the redshift of a quasar from five-band (low spectral resolution) photometric data, the so called "photo-z" problem. Our method shows that tools from machine learning and Bayesian statistics allow us to leverage multiple resolutions of information to make accurate predictions with well-characterized uncertainties.

## 1   Introduction

Enormous amounts of astronomical data are collected by a range of instruments at multiple spectral resolutions, providing information about billions of sources of light in the observable universe [1, 10]. Among these data are measurements of the spectral energy distributions (SEDs) of sources of light (e.g. stars, galaxies, and quasars). The SED describes the distribution of energy radiated by a source over the spectrum of wavelengths or photon energy levels. SEDs are of interesting because they convey information about a source's physical properties, including type, chemical composition, and redshift, which will be an estimand of interest in this work.

The SED can be thought of as a latent function of which we can only obtain noisy measurements. Measurements of SEDs, however, are produced by instruments at widely varying spectral resolutions – some instruments measure many wavelengths simultaneously (spectroscopy), while others

---

[*] http://people.seas.harvard.edu/~acm/
[†] http://people.seas.harvard.edu/~rpa/

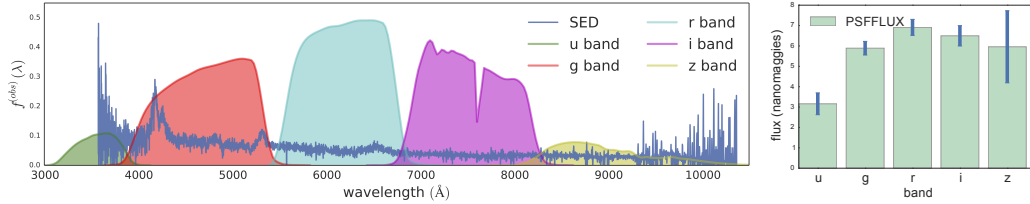

Figure 1: Left: example of a BOSS-measured quasar SED with SDSS band filters, $S_b(\lambda)$, $b \in \{u, g, r, i, z\}$, overlaid. Right: the same quasar's photometrically measured band fluxes. Spectroscopic measurements include noisy samples at thousands of wavelengths, whereas SDSS photometric fluxes reflect the (weighted) response over a large range of wavelengths.

average over large swaths of the energy spectrum and report a low dimensional summary (photometry). Spectroscopic data describe a source's SED in finer detail than broadband photometric data. For example, the Baryonic Oscillation Spectroscopic Survey [5] measures SED samples at over four thousand wavelengths between 3,500 and 10,500 Å. In contrast, the Sloan Digital Sky Survey (SDSS) [1] collects spectral information in only 5 broad spectral bins by using broadband filters (called $u, g, r, i$, and $z$), but at a much higher *spatial* resolution. Photometric preprocessing models can then aggregate pixel information into five band-specific fluxes and their uncertainties [17], reflecting the weighted average response over a large range of the wavelength spectrum. The two methods of spectral information collection are graphically compared in Figure 1.

Despite carrying less spectral information, broadband photometry is more widely available and exists for a larger number of sources than spectroscopic measurements. This work develops a method for inferring physical properties sources by jointly modeling spectroscopic and photometric data. One use of our model is to measure the redshift of quasars for which we only have photometric observations. Redshift is a phenomenon in which the observed SED of a source of light is stretched toward longer (redder) wavelengths. This effect is due to a combination of radial velocity with respect to the observer and the expansion of the universe (termed *cosmological redshift*) [8, 7]. Quasars, or quasi-stellar radio sources, are extremely distant and energetic sources of electromagnetic radiation that can exhibit high redshift [16]. Accurate estimates and uncertainties of redshift measurements from photometry have the potential to guide the use of higher spectral resolution instruments to study sources of interest. Furthermore, accurate photometric models can aid the automation of identifying source types and estimating physical characteristics of faintly observed sources in large photometric surveys [14].

To jointly describe both resolutions of data, we directly model a quasar's latent SED and the process by which it generates spectroscopic and photometric observations. Representing a quasar's SED as a latent random measure, we describe a Bayesian inference procedure to compute the marginal probability distribution of a quasar's redshift given observed photometric fluxes and their uncertainties. The following section provides relevant application and statistical background. Section 3 describes our probabilistic model of SEDs and broadband photometric measurements. Section 4 outlines our MCMC-based inference method for efficiently computing statistics of the posterior distribution. Section 5 presents redshift and SED predictions from photometric measurements, among other model summaries, and a quantitative comparison between our method and two existing "photo-z". We conclude with a discussion of directions for future work.

## 2 Background

The SEDs of most stars are roughly approximated by Planck's law for black body radiators and stellar atmosphere models [6]. Quasars, on the other hand, have complicated SEDs characterized by some salient features, such as the Lyman-$\alpha$ forest, which is the absorption of light at many wavelengths from neutral hydrogen gas between the earth and the quasar [19]. One of the most interesting properties of quasars (and galaxies) conveyed by the SED is redshift, which gives us insight into an object's distance and age. Redshift affects our observation of SEDs by "stretching" the wavelengths, $\lambda \in \Lambda$, of the quasar's *rest frame* SED, skewing toward longer (redder) wavelengths. Denoting the *rest frame* SED of a quasar $n$ as a function, $f_n^{(\text{rest})} : \Lambda \to \mathbb{R}_+$, the effect of redshift with value $z_n$

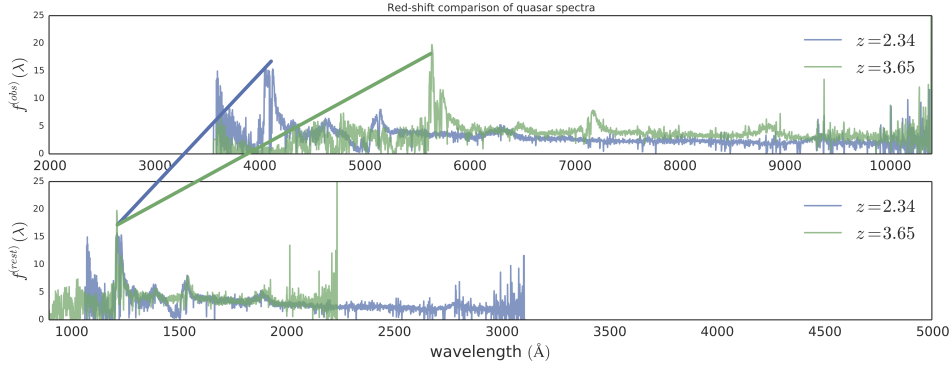

Figure 2: Spectroscopic measurements of multiple quasars at different redshifts, $z$. The upper graph depicts the sample spectrograph in the observation frame, intuitively thought of as "stretched" by a factor $(1 + z)$. The lower figure depicts the "de-redshifted" (rest frame) version of the same quasar spectra, The two lines show the corresponding locations of the characteristic peak in each reference frame. Note that the $x$-axis has been changed to ease the visualization - the transformation is much more dramatic. The appearance of translation is due to missing data; we don't observe SED samples outside the range 3,500-10,500 Å.

(typically between 0 and 7) on the *observation-frame* SED is described by the relationship

$$f_n^{(\text{obs})}(\lambda) = f_n^{(\text{rest})}\left(\frac{\lambda}{1 + z_n}\right).$$ (1)

Some observed quasar spectra and their "de-redshifted" rest frame spectra are depicted in Figure 2.

## 3 Model

This section describes our probabilistic model of spectroscopic and photometric observations.

**Spectroscopic flux model** The SED of a quasar is a non-negative function $f : \Lambda \to \mathbb{R}_+$, where $\Lambda$ denotes the range of wavelengths and $\mathbb{R}_+$ are non-negative real numbers representing flux density. Our model specifies a quasar's *rest frame* SED as a latent random function. Quasar SEDs are highly structured, and we model this structure by imposing the assumption that each SED is a convex mixture of $K$ latent, positive basis functions. The model assumes there are a small number ($K$) of latent features or characteristics and that each quasar can be described by a short vector of mixing weights over these features.

We place a normalized log-Gaussian process prior on each of these basis functions (described in supplementary material). The generative procedure for quasar spectra begins with a shared basis

$$\beta_k(\cdot) \stackrel{\text{iid}}{\sim} \mathcal{GP}(0, K_\theta), \ k = 1, \dots, K, \qquad B_k(\cdot) = \frac{\exp(\beta_k(\cdot))}{\int_\Lambda \exp(\beta_k(\lambda))\, d\lambda},$$ (2)

where $K_\theta$ is the kernel and $B_k$ is the exponentiated and normalized version of $\beta_k$. For each quasar $n$,

$$\mathbf{w}_n \sim p(\mathbf{w})\,, \text{ s.t. } \sum_{w_k} w_k = 1, \qquad m_n \sim p(m)\,, \text{ s.t. } m_n > 0, \qquad z_n \sim p(z),$$ (3)

where $\mathbf{w}_n$ mixes over the latent types, $m_n$ is the apparent brightness, $z_n$ is the quasar's redshift, and distributions $p(\mathbf{w})$, $p(m)$, and $p(z)$ are priors to be specified later. As each positive SED basis function, $B_k$, is normalized to integrate to one, and each quasar's weight vector $\mathbf{w}_n$ also sums to one, the latent *normalized* SED is then constructed as

$$f_n^{(\text{rest})}(\cdot) = \sum_k w_{n,k} B_k(\cdot)$$ (4)

and we define the unnormalized SED $\tilde{f}_n^{(\text{rest})}(\cdot) \equiv m_n \cdot f_n^{(\text{rest})}(\cdot)$. This parameterization admits the interpretation of $f_n^{(\text{rest})}(\cdot)$ as a probability density scaled by $m_n$. This interpretation allows us to

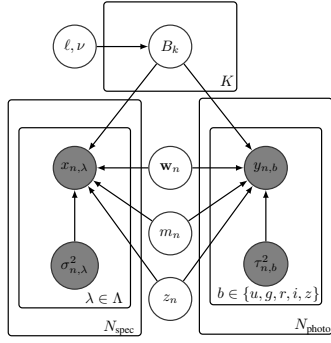

Figure 3: Graphical model representation of the joint photometry and spectroscopy model. The left shaded variables represent spectroscopically measured samples and their variances. The right shaded variables represent photometrically measured fluxes and their variances. The upper box represents the latent basis, with GP prior parameters $\ell$ and $\nu$. Note that $N_{\text{spec}} + N_{\text{photo}}$ replicates of $\mathbf{w}_n, m_n$ and $z_n$ are instantiated.

separate out the apparent brightness, which is a function of distance and overall luminosity, from the SED itself, which carries information pertinent to the estimand of interest, redshift.

For each quasar with spectroscopic data, we observe noisy samples of the redshifted and scaled spectral energy distribution at a grid of $P$ wavelengths $\lambda \in \{\lambda_1, \ldots, \lambda_P\}$. For quasar $n$, our *observation frame* samples are conditionally distributed as

$$x_{n,\lambda}|z_n, \mathbf{w}_n, \{B_k\} \overset{\text{ind}}{\sim} \mathcal{N}\left(\tilde{f}_n^{(\text{rest})}\left(\frac{\lambda}{1+z_n}\right), \sigma_{n,\lambda}^2\right) \tag{5}$$

where $\sigma_{n,\lambda}^2$ is known measurement variance from the instruments used to make the observations. The BOSS spectra (and our rest frame basis) are stored in units $10^{-17} \cdot \text{erg} \cdot \text{cm}^{-2} \cdot \text{s}^{-1} \cdot \mathring{\text{A}}^{-1}$.

**Photometric flux model**   Photometric data summarize the amount of energy observed over a large swath of the wavelength spectrum. Roughly, a photometric flux measures (proportionally) the number of photons recorded by the instrument over the duration of an exposure, filtered by a band-specific sensitivity curve. We express flux in nanomaggies [15]. Photometric fluxes and measurement error derived from broadband imagery have been computed directly from pixels [17]. For each quasar $n$, SDSS photometric data are measured in five bands, $b \in \{u, g, r, i, z\}$, yielding a vector of five flux values and their variances, $\mathbf{y}_n$ and $\tau_{n,b}^2$. Each band, $b$, measures photon observations at each wavelength in proportion to a known filter sensitivity, $S_b(\lambda)$. The filter sensitivities for the SDSS $ugriz$ bands are depicted in Figure 1, with an example observation frame quasar SED overlaid. The actual measured fluxes can be computed by integrating the full object's spectrum, $m_n \cdot f_n^{(\text{obs})}(\lambda)$ against the filters. For a band $b \in \{u, g, r, i, z\}$

$$\mu_b(f_n^{(\text{rest})}, z_n) = \int f_n^{(\text{obs})}(\lambda) \, S_b(\lambda) \, C(\lambda) \, d\lambda, \tag{6}$$

where $C(\lambda)$ is a conversion factor to go from the units of $f_n(\lambda)$ to nanomaggies (details of this conversion are available in the supplementary material). The function $\mu_b$ takes in a rest frame SED, a redshift ($z$) and maps it to the observed $b$-band specific flux. The results of this projection onto SDSS bands are modeled as independent Gaussian random variables with known variance

$$y_{n,b} \mid f_n^{(\text{rest})}, z_n \overset{\text{ind}}{\sim} \mathcal{N}(\mu_b(f_n^{(\text{rest})}, z_n), \tau_{n,b}^2). \tag{7}$$

Conditioned on the basis, $B = \{B_k\}$, we can represent $f_n^{(\text{rest})}$ with a low-dimensional vector. Note that $f_n^{(\text{rest})}$ is a function of $\mathbf{w}_n, z_n, m_n$, and $B$ (see Equation 4), so we can think of $\mu_b$ as a function of $\mathbf{w}_n, z_n, m_n$, and $B$. We overload notation, and re-write the conditional likelihood of photometric observations as

$$y_{n,b} \mid \mathbf{w}_n, z_n, m_n, B \sim \mathcal{N}(\mu_b(\mathbf{w}_n, z_n, m_n, B), \tau_{n,b}^2). \tag{8}$$

Intuitively, what gives us statistical traction in inferring the posterior distribution over $z_n$ is the structure learned in the latent basis, $B$, and weights $w$, i.e., the features that correspond to distinguishing bumps and dips in the SED.

**Note on priors**   For photometric weight and redshift inference, we use a flat prior on $z_n \in [0, 8]$, and empirically derived priors for $m_n$ and $w_n$, from the sample of spectroscopically measured sources. Choice of priors is described in the supplementary material.

# 4 Inference

**Basis estimation**    For computational tractability, we first compute a maximum a posteriori (MAP) estimate of the basis, $B_{\text{map}}$ to condition on. Using the spectroscopic data, $\{x_{n,\lambda}, \sigma_{n,\lambda}^2, z_n\}$, we compute a discretized MAP estimate of $\{B_k\}$ by directly optimizing the unnormalized (log) posterior implied by the likelihood in Equation 5, the GP prior over $B$, and diffuse priors over $\mathbf{w}_n$ and $m_n$,

$$p\left(\{\mathbf{w}_n, m_n\}, \{B_k\}|\{x_{n,\lambda}, \sigma_{n,\lambda}^2, z_n\}\right) \propto \prod_{n=1}^{N} p(x_{n,\lambda}|z_n, \mathbf{w}_n, m_n, \{B_k\})p(\{B_k\})p(\mathbf{w}_n)p(m_n). \tag{9}$$

We use gradient descent with momentum and LBFGS [12] directly on the parameters $\beta_k, \omega_{n,k}$, and $\log(m_n)$ for the $N_{spec}$ spectroscopically measured quasars. Gradients were automatically computed using `autograd` [9]. Following [18], we first resample the observed spectra into a common rest frame grid, $\lambda_0 = (\lambda_{0,1}, \ldots, \lambda_{0,V})$, easing computation of the likelihood. We note that although our model places a full distribution over $B_k$, efficiently integrating out those parameters is left for future work.

**Sampling $\mathbf{w}_n, m_n$, and $z_n$**    The Bayesian "photo-z" task requires that we compute posterior marginal distributions of $z$, integrating out $\mathbf{w}$, and $m$. To compute these distributions, we construct a Markov chain over the state space including $z$, $\mathbf{w}$, and $m$ that leaves the target posterior distribution invariant. We treat the inference problem for each photometrically measured quasar, $\mathbf{y}_n$, independently. Conditioned on a basis $B_k, k = 1, \ldots, K$, our goal is to draw posterior samples of $\mathbf{w}_n, m_n$ and $z_n$ for each $n$. The unnormalized posterior can be expressed

$$p(\mathbf{w}_n, m_n, z_n|\mathbf{y}_n, B) \propto p(\mathbf{y}_n|\mathbf{w}_n, m_n, z_n, B)p(\mathbf{w}_n, m_n, z_n) \tag{10}$$

where the left likelihood term is defined in Equation 8. Note that due to analytic intractability, we numerically integrate expressions involving $\int_\Lambda f_n^{(obs)}(\lambda)d\lambda$ and $S_b(\lambda)$. Because the observation $\mathbf{y}_n$ can often be well explained by various redshifts and weight settings, the resulting marginal posterior, $p(z_n|\mathbf{X}, \mathbf{y}_n, B)$, is often multi-modal, with regions of near zero probability between modes. Intuitively, this is due to the information loss in the SED-to-photometric flux integration step.

This multi-modal property is problematic for many standard MCMC techniques. Single chain MCMC methods have to jump between modes or travel through a region of near-zero probability, resulting in slow mixing. To combat this effect, we use parallel tempering [4], a method that is well-suited to constructing Markov chains on multi-modal distributions. Parallel tempering instantiates $C$ independent chains, each sampling from the target distribution raised to an inverse temperature. Given a target distribution, $\pi(x)$, the constructed chains sample $\pi_c(x) \propto \pi(x)^{1/T_c}$, where $T_c$ controls how "hot" (i.e., how close to uniform) each chain is. At each iteration, swaps between chains are proposed and accepted with a standard Metropolis-Hastings acceptance probability

$$\Pr(\text{accept swap } c, c') = \frac{\pi_c(x_{c'})\pi_{c'}(x_c)}{\pi_c(x_c)\pi_{c'}(x_{c'})}. \tag{11}$$

Within each chain, we use component-wise slice sampling [11] to generate samples that leave each chain's distribution invariant. Slice-sampling is a (relatively) tuning-free MCMC method, a convenient property when sampling from thousands of independent posteriors. We found parallel tempering to be essential for convincing posterior simulations. MCMC diagnostics and comparisons to single-chain samplers are available in the supplemental material.

# 5 Experiments and Results

We conduct three experiments to test our model, where each experiment measures redshift predictive accuracy for a different train/test split of spectroscopically measured quasars from the DR10QSO dataset [13] with confirmed redshifts in the range $z \in (.01, 5.85)$. Our experiments split train/test in the following ways: (i) randomly, (ii) by $r$-band fluxes, (iii) by redshift values. In split (ii), we train on the brightest 90% of quasars, and test on a subset of the remaining. Split (iii) takes the lowest 85% of quasars as training data, and a subset of the brightest 15% as test cases. Splits (ii)

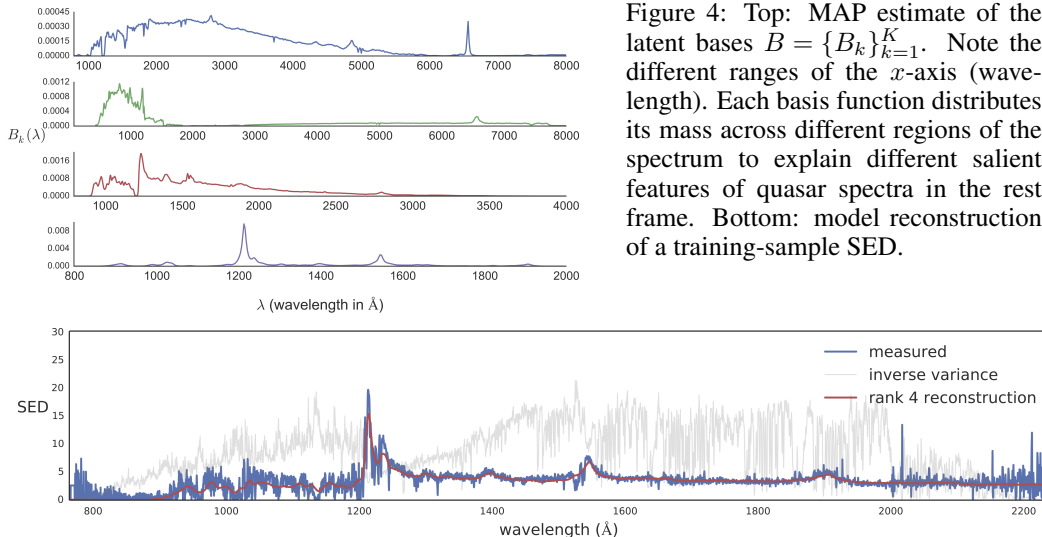

Figure 4: Top: MAP estimate of the latent bases $B = \{B_k\}_{k=1}^K$. Note the different ranges of the $x$-axis (wavelength). Each basis function distributes its mass across different regions of the spectrum to explain different salient features of quasar spectra in the rest frame. Bottom: model reconstruction of a training-sample SED.

and (iii) are intended to test the method's robustness to different training and testing distributions, mimicking the discovery of fainter and farther sources. For each split, we find a MAP estimate of the basis, $B_1, \ldots, B_K$, and weights, $\mathbf{w}_n$ to use as a prior for photometric inference. For computational purposes, we limit our training sample to a random subsample of 2,000 quasars. The following sections outline the resulting model fit and inferred SEDs and redshifts.

**Basis validation**    We examined multiple choices of $K$ using out of sample likelihood on a validation set. In the following experiments we set $K = 4$, which balances generalizability and computational tradeoffs. Discussion of this validation is provided in the supplementary material.

**SED Basis**    We depict a MAP estimate of $B_1, \ldots, B_K$ in Figure 4. Our basis decomposition enjoys the benefit of physical interpretability due to our density-estimate formulation of the problem. Basis $B_4$ places mass on the Lyman-$\alpha$ peak around 1,216 Å, allowing the model to capture the co-occurrence of more peaked SEDs with a bump around 1,550 Å. Basis $B_1$ captures the H-$\alpha$ emission line at around 6,500 Å. Because of the flexible nonparametric priors on $B_k$ our model is able to automatically learn these features from data. The positivity of the basis and weights distinguishes our model from PCA-based methods, which sacrifice physical interpretability.

**Photometric measurements**    For each test quasar, we construct an 8-chain parallel tempering sampler and run for 8,000 iterations, and discard the first 4,000 samples as burn-in. Given posterior samples of $z_n$, we take the posterior mean as a point estimate. Figure 5 compares the posterior mean to spectroscopic measurements (for three different data-split experiments), where the gray lines denote posterior sample quantiles. In general there is a strong correspondence between spectroscopically measured redshift and our posterior estimate. In cases where the posterior mean is off, our distribution often covers the spectroscopically confirmed value with probability mass. This is clear upon inspection of posterior marginal distributions that exhibit extreme multi-modal behavior. To combat this multi-modality, it is necessary to inject the model with more information to eliminate plausible hypotheses; this information could come from another measurement (e.g., a new photometric band), or from structured prior knowledge over the relationship between $z_n, \mathbf{w}_n$, and $m_n$. Our method simply fits a mixture of Gaussians to the spectroscopically measured $\mathbf{w}_n, m_n$ sample to formulate a prior distribution. However, incorporating dependencies between $z_n, \mathbf{w}_n$ and $m_n$, similar to the XDQSOz technique, will be incorporated in future work.

## 5.1   Comparisons

We compare the performance of our redshift estimator with two recent photometric redshift estimators, XDQSOz [2] and a neural network [3]. The method in [2] is a conditional density estimator that discretizes the range of one flux band (the $i$-band) and fits a mixture of Gaussians to the joint distribution over the remaining fluxes and redshifts. One disadvantage to this approach is there there

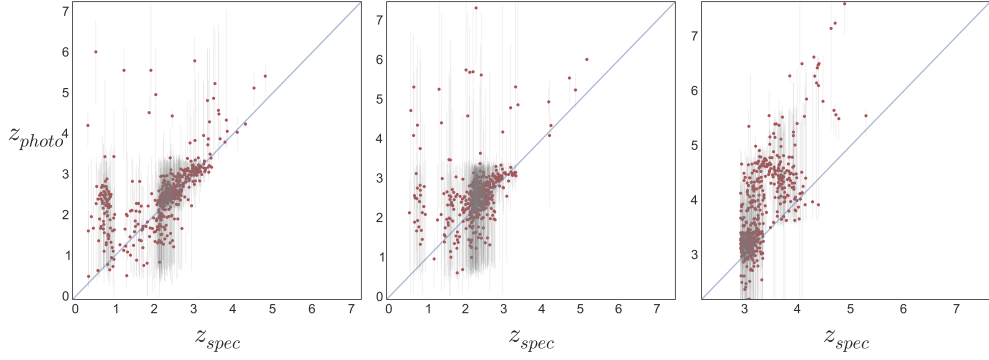

Figure 5: Comparison of spectroscopically ($x$-axis) and photometrically ($y$-axis) measured redshifts from the SED model for three different data splits. The left reflects a random selection of 4,000 quasars from the DR10QSO dataset. The right graph reflects a selection of 4,000 test quasars from the upper 15% ($z_{cutoff} \approx 2.7$), where all training was done on lower redshifts. The red estimates are posterior means.

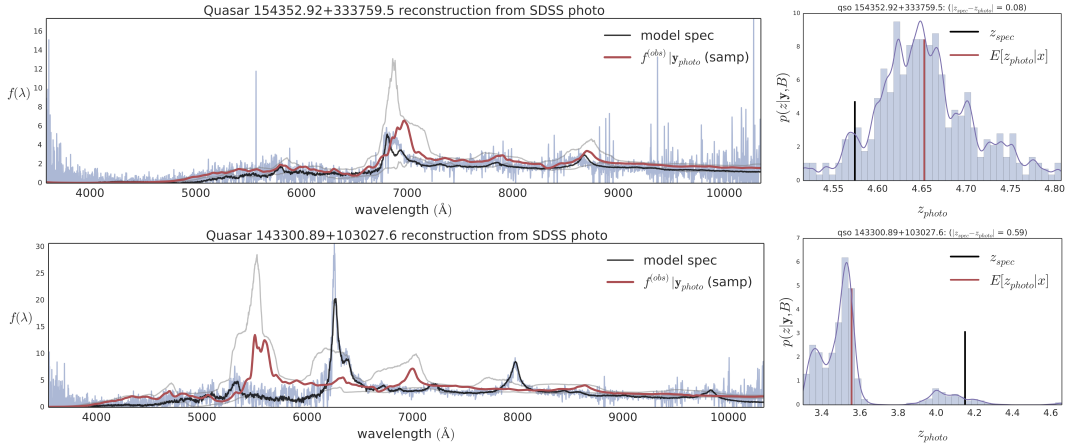

Figure 6: Left: inferred SEDs from photometric data. The black line is a smoothed approximation to the "true" SED using information from the full spectral data. The red line is a sample from the posterior, $f_n^{(obs)}(\lambda)|\mathbf{X}, \mathbf{y}_n, B$, which imputes the entire SED from only five flux measurements. Note that the bottom sample is from the left mode, which under-predicts redshift. Right: corresponding posterior predictive distributions, $p(z_n|\mathbf{X}, \mathbf{y}_n, B)$. The black line marks the spectroscopically confirmed redshift; the red line marks the posterior mean. Note the difference in scale of the $x$-axis.

is no physical significance to the mixture of Gaussians, and no model of the latent SED. Furthermore, the original method trains and tests the model on a pre-specified range of $i$-magnitudes, which is problematic when predicting redshifts on much brighter or dimmer stars. The regression approach from [3] employs a neural network with two hidden layers, and the SDSS fluxes as inputs. More features (e.g., more photometric bands) can be incorporated into all models, but we limit our experiments to the five SDSS bands for the sake of comparison. Further detail on these two methods and a broader review of "photo-z" approaches are available in the supplementary material.

**Average error and test distribution** We compute mean absolute error (MAE), mean absolute percentage error (MAPE), and root mean square error (RMSE) to measure predictive performance. Table 1 compares prediction errors for the three different approaches (XD, NN, Spec). Our experiments show that accurate redshift measurements are attainable even when the distribution of training set is different from test set by directly modeling the SED itself. Our method dramatically outperforms [2] and [3] in split (iii), particularly for very high redshift fluxes. We also note that our training set is derived from only 2,000 examples, whereas the training set for XDQSOz and the neural network were $\approx$ 80,000 quasars and 50,000 quasars, respectively. This shortcoming can be overcome with more sophisticated inference techniques for the non-negative basis. Despite this, the

| split | MAE | | | MAPE | | | RMSE | | |
|---|---|---|---|---|---|---|---|---|---|
| | XD | NN | Spec | XD | NN | Spec | XD | NN | Spec |
| random (all) | **0.359** | 0.773 | 0.485 | **0.293** | 0.533 | 0.430 | **0.519** | 0.974 | 0.808 |
| flux (all) | **0.308** | 0.483 | 0.497 | **0.188** | 0.283 | 0.339 | **0.461** | 0.660 | 0.886 |
| redshift (all) | 0.841 | 0.736 | **0.619** | 0.237 | 0.214 | **0.183** | 1.189 | 0.923 | **0.831** |
| random ($z > 2.35$) | **0.247** | 0.530 | 0.255 | **0.091** | 0.183 | 0.092 | **0.347** | 0.673 | 0.421 |
| flux ($z > 2.33$) | **0.292** | 0.399 | 0.326 | **0.108** | 0.143 | 0.124 | **0.421** | 0.550 | 0.531 |
| redshift ($z > 3.20$) | 1.327 | 1.149 | **0.806** | 0.357 | 0.317 | **0.226** | 1.623 | 1.306 | **0.997** |
| random ($z > 3.11$) | **0.171** | 0.418 | 0.289 | **0.050** | 0.117 | 0.082 | **0.278** | 0.540 | 0.529 |
| flux ($z > 2.86$) | 0.373 | 0.493 | **0.334** | 0.112 | 0.144 | **0.103** | **0.606** | 0.693 | 0.643 |
| redshift ($z > 3.80$) | 2.389 | 2.348 | **0.829** | 0.582 | 0.569 | **0.198** | 2.504 | 2.405 | **1.108** |

Table 1: Prediction error for three train-test splits, (i) random, (ii) flux-based, (iii) redshift-based, corresponding to XDQSOz [2] (XD), the neural network approach [3] (NN), our SED-based model (Spec). The middle and lowest sections correspond to test redshifts in the upper 50% and 10%, respectively. The XDQSOz and NN models were trained on (roughly) 80,000 and 50,000 example quasars, respectively, while the Spec models were trained on 2,000.

SED-based predictions are comparable. Additionally, because we are directly modeling the latent SED, our method admits a posterior estimate of the entire SED. Figure 6 displays posterior SED samples and their corresponding redshift marginals for test-set quasars inferred from only SDSS photometric measurements.

## 6 Discussion

We have presented a generative model of two sources of information at very different spectral resolutions to form an estimate of the latent spectral energy distribution of quasars. We also described an efficient MCMC-based inference algorithm for computing posterior statistics given photometric observations. Our model accurately predicts and characterizes uncertainty about redshifts from only photometric observations and a small number of separate spectroscopic examples. Moreover, we showed that we can make reasonable estimates of the unobserved SED itself, from which we can make inferences about other physical properties informed by the full SED.

We see multiple avenues of future work. Firstly, we can extend the model of SEDs to incorporate more expert knowledge. One such augmentation would include a fixed collection of features, curated by an expert, corresponding to physical properties already known about a class of sources. Furthermore, we can also extend our model to directly incorporate photometric pixel observations, as opposed to preprocessed flux measurements. Secondly, we note that our method is more more computationally burdensome than XDQSOz and the neural network approach. Another avenue of future work is to find accurate approximations of these posterior distributions that are cheaper to compute. Lastly, we can extend our methodology to galaxies, whose SEDs can be quite complicated. Galaxy observations have spatial extent, complicating their SEDs. The combination of SED and spatial appearance modeling and computationally efficient inference procedures is a promising route toward the automatic characterization of millions of sources from the enormous amounts of data available in massive photometric surveys.

## Acknowledgments

The authors would like to thank Matthew Hoffman and members of the HIPS lab for helpful discussions. This work is supported by the Applied Mathematics Program within the Office of Science Advanced Scientific Computing Research of the U.S. Department of Energy under contract No. DE-AC02-05CH11231. This work used resources of the National Energy Research Scientific Computing Center (NERSC). We would like to thank Tina Butler, Tina Declerck and Yushu Yao for their assistance.

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
