[Supplementary Material · mwa_quasar_SI.pdf]

# A Gaussian Process Model of Quasar Spectral Energy Distributions: Supplementary Material

**Andrew Miller**[*] **, Albert Wu**
School of Engineering and Applied Sciences
Harvard University
acm@seas.harvard.edu

**Jeffrey Regier,  Jon McAuliffe**
Department of Statistics
University of California, Berkeley
{jeff, jon}@stat.berkeley.edu

**Dustin Lang**
McWilliams Center for Cosmology
Carnegie Mellon University
dstn@cmu.edu

**Prabhat,  David Schlegel**
Lawrence Berkeley National Laboratory
{prabhat, djschlegel}@lbl.gov

**Ryan Adams** [†]
Harvard University
Cambridge, MA 02138
rpa@seas.harvard.edu

## 1   Prior Specification

### 1.1   Gaussian process priors

We use a Gaussian process prior to model the complicated shape of quasar SEDs. GPs allow us to flexibly encode our prior beliefs about the structure and shape of a function. A Gaussian process (GP) is a stochastic process, $f : \mathcal{X} \to \mathbb{R}$, such that any finite collection of random variables, $f(x_1), \ldots, f(x_N) \in \mathbb{R}$, is distributed according to a multivariate normal distribution. GPs are frequently used as priors over unknown functions, $f$, where the random variables $f(x_1), \ldots, f(x_N)$ correspond to evaluations of the function at inputs $x_1, \ldots, x_N \in \mathcal{X}$. The prior covariance between any two outputs, $f(x_i)$ and $f(x_j)$, encodes prior beliefs about the function $f$; carefully chosen covariance functions can encode beliefs about a wide range of properties, including smoothness and periodicity.

Throughout this paper we use the the Matérn [8] covariance function

$$k_{\text{Matern}}(r) = \frac{2^{1-\nu}}{\Gamma(\nu)} \left( \frac{\sqrt{\nu}r}{\ell} \right)^{\nu} K_{\nu} \left( \frac{\sqrt{2\nu}r}{\ell} \right) \tag{1}$$

where $r = |x_1 - x_2|$, and $K_{\nu}$ is a modified Bessel function. The parameter $\nu$ controls the smoothness and $\ell$ is the length scale of the function. We choose the Matérn covariance function for its ability to trade off between smooth and flexible sample paths. Because this covariance is strictly a function of the distance between two points in the space $\mathcal{X}$, it is said to be stationary. See [9] for a thorough treatment of Gaussian processes in machine learning.

---

[*]http://people.seas.harvard.edu/~acm/
[†]http://people.seas.harvard.edu/~rpa/

Our prior over basis $B_k$ is a normalized log-Gaussian process, which is distributed

$$\beta_k \sim \mathcal{GP}(0, K_\theta) \tag{2}$$

$$B_k = \frac{\exp(\beta_k)}{\int \exp(\beta_k(\lambda))d\lambda} \, . \tag{3}$$

We use the above construction and resort to numerical integration to normalize the one-dimensional basis functions in our implementation.

## 1.2  Weight and Redshift Priors

In the photometric inference task, we define priors over $\mathbf{w}$, $m$, and $z$ before computing posterior samples. We place a flat prior over $z \in [0, 8]$. For the vector of weights, $\mathbf{w}_k$, we use the statistics of the spectroscopic sample to determine a prior. That is, we fit the non-negative basis and loadings model to $N_{spec} \approx 2,000$ full BOSS spectra, which gives us basis loadings for each spectroscopically measured quasar, $\mathbf{w}_n$ for $n = 1, \ldots, N_{spec}$. We use this sample as a prior distribution over weights. Specifically, we fit a mixture of Gaussians to the logit-space variables, $\omega_1, \ldots, \omega_{K-1}$, defined

$$\omega_k = \log(w_k) - \log(w_K) \qquad w_k = \frac{\exp(\omega_k)}{1 + \sum_{k'} \exp(\omega_{k'})} \, . \tag{4}$$

We determine the number of mixture components by comparing likelihoods on a held-out validation set. We then use this prior on the $\omega$ weights for simulating posterior samples in the photometric model.

## 2  SED to Photometric Band Conversion

Given a spectral energy distribution, $f(\lambda)$, expressed in units $10^{-17} \cdot \mathrm{erg} \cdot \mathrm{cm}^{-2} \cdot \mathrm{s}^{-1} \cdot \text{Å}^{-1}$, we can compute the expected flux observation corresponding to an SDSS photometric band $b$. This flux can be computed by computed by integrating the full object's spectrum, $m_n \cdot f_n^{(\mathrm{obs})}(\lambda)$ against the filters. For a band $b \in \{u, g, r, i, z\}$

$$\mu_b(f_n^{(\mathrm{rest})}, z_n) = \int f_n^{(\mathrm{obs})}(\lambda)\, S_b(\lambda)\, C(\lambda)\, d\lambda, \quad C(\lambda) = \frac{1}{\int S_b(\lambda)d\lambda} \frac{\lambda^2}{c} 10^{(48.6 - 2.5 \cdot 17 + 22.5)/2.5} \tag{5}$$

where $C(\lambda)$ is a conversion factor to go from the units of $f_n(\lambda)$ to nanomaggies. The values in the expression above correspond to arbitrary zero-points and constants used in astronomical units: $48.6$ is a zero-point for an astronomical measurement called AB magnitude, $17$ is used to match fluxes which are in units of $10^{-17}$ ergs, $22.5$ and $2.5$ are constants used to convert from logarithmic to linear units of brightness, and $c$ is the speed of light in $\text{Å}/s$.

## 3  Related work

Many machine learning and statistical methods have been applied to the "photo-$z$" problem. The review by [12] divides "photo-$z$" methods into two categories, empirical and template-fitting. Empirical methods are often discriminative, regression-based approaches, whereas template-fitting methods are often SED model-based approaches.

A recently proposed empirical method uses a multi-layer perceptron with a combination of photometric datasets, including SDSS, to compute a regression function for redshift [3]. This method, while efficient and accurate in mean error, does not characterize the uncertainty of the estimated redshift, nor does it admit any physically interpretable estimate of the SED and quasar type.

Template-based approaches use information derived from spectroscopy to assist redshift predictions. [5] and [10] present a clustering algorithm for reconstructing quasar SEDs from photometric observations and templates derived from noisy spectroscopic measurements. Their method clusters quasars into individual template categories using a $K$-means-like update, relying on weighted template averages of noisy SED measurements and unique cluster assignments that do not express

uncertainty over quasar "types". Instead of a pure clustering method, we use a non-negative factor analysis-type model to represent the latent structure of quasar SEDs and a continuous, low-dimensional range of quasar types. Furthermore, our method presents a fully probabilistic model of SEDs and a Bayesian inference procedure that integrates out uncertainty over latent variables, including quasar type and apparent brightness, to measure redshift.

Other models blur the line between regression-based and generative models. [2] develops the XDQSOz method to use a large dataset of astronomical objects to simultaneously infer redshift and classify quasars. They model the joint distribution over object type, fluxes, and redshift. However, their method does not model SEDs themselves, nor the SED-to-flux generative process.

For further reference, [1] presents a thorough summary of Bayesian methods for photometric redshift estimation from spectral templates. [4] unifies template-based and regression-based approaches into a single probabilistic framework.

## 3.1 Comparison

The method in [2], XDQSOz, involves binning quasars by $i$-band magnitude and then, within each bin, fitting a mixture of Gaussians to the joint distribution of redshift and the relative magnitudes of the other bands. In our analysis, we binned the $i$-magnitude into bins of width 0.2 and then fit a mixture of Gaussians to the joint distribution in each bin, where the number of Gaussians is chosen through validation. We can calculate a marginal distribution over redshift given the $ugriz$ bands, and we use both the mean and MLE of this posterior as point predictions for comparison.

A disadvantage of the XDQSOz approach is there there is no physical significance to the mixture of Gaussians. Furthermore, the original method trains and tests the model on a pre-specified range of $i$-magnitudes, which is problematic if we want to predict the redshifts of much brighter or much dimmer stars. For the sake of a fair comparison, we bin the out-of-range $i$-magnitudes into the lowest and highest bins appropriately in our experiments.

The method from [3] employs a neural network with two hidden layers. Given that the number of inputs is $N$, the first hidden layer has $(2N + 1)$ outputs and the second hidden layer has anywhere from 1 to $(N - 1)$ outputs. Tuning the number of hidden nodes through validation, we used a neural network with 5 nodes in the input layer, for each of the five $ugriz$ bands, 11 nodes in the first hidden layer, 2 nodes in the second hidden layer, and a single output node. We used the implementation from [11] for our neural network experiments.

## 4 Validating $K$

To validate $K$, we examined the predictive likelihood of the SED model on a set of held-out spectroscopy data. We fit models for $K = 2, 4, 6, 8, 10, 15$ on 2,000 training quasar spectra and computed the predictive likelihood on a set of 2,000 out of sample quasar spectra. We found that increasing $K$ improved generalization all the way up to $K = 15$, where it becomes computationally burdensome to compute maximum likelihood estimates without more sophisticated optimization techniques. The distribution of test likelihoods are depicted in Figure 1.

However, better generalization for spectroscopy data does not immediately translate into better generalization for photometry data. We also examined two $K$ values for photometric redshift prediction, $K = 4$ and $K = 6$. When inferring redshift and weights from a quasar's photometric data (i.e. 5 real valued flux measurements and their uncertainties), we found that $K = 6$ yielded less accurate predictions in terms of mean absolute error and mean absolute percentage error. Our hypothesis is that without a structured prior that relates $z$ and $\mathbf{w}$ (similar to XDQSOz), higher values of $K$ make MCMC integration more difficult and underdetermined. In future work we will specify a model for the interactions between $\mathbf{w}$, $z$ and $m$, including potential low-rank structure among the population $\mathbf{w}$ matrix. With more functional bases, it seems necessary to enforce more structure on the weights to yield reasonable estimates in the low-information photometric regime.

Figure 1: Distribution of predictive likelihoods for held-out spectroscopy data. We trained a basis model on values $K = 2, 4, 6, 8, 10, 15$, and test out of sample predictive likelihood. We see that extra bases improve predictive performance for spectroscopy data, however this does not directly translate to predictive performance for photometry data.

|  | PT+SS | SS |
|---|---|---|
| $\hat{r}$ | 1.05 | 4.00 |
| $N_{eff}$ | 56.04 | 4.33 |
| ESS | 972.74 | 102.57 |

Table 1: Comparison of MCMC diagnostic statistics. $\hat{r}$ refers to the potential scale reduction factor [7]. $N_{eff}$ is a measure of effective sample size based on between and within-chain variance [6]. ESS is a measurement of effectively independent samples based on autocorrelation. We found parallel-tempering to consistently outperform a single chain, especially in our "many-posterior" tuning-free setting.

## 5    MCMC Comparison: Parallel-Tempering vs. Single Chain

This section addresses the choice of slice-sampling within parallel-tempering (PT-SS) versus a single chain slice-sampler (SS). As mentioned in the main paper, our single-chain method of choice is slice-sampling due to its ease of implementation and relatively hands-off tuning. Conditioned on a map basis, $B_{mle}$, and photometric fluxes, $\{y, \tau^2\}_{ugriz}$, the posterior over basis loadings and redshift is often multimodal. This intuitively makes sense - because the photometric collection process washes away most of the detailed spectral information, multiple SED hypotheses may suitably explain photometric observations.

This phenomenon was observed when simulating posterior samples. Figure 2 compares the paths of four single-chain slice samplers to four parallel tempering slice samplers (with 8 temperature levels each). Single-chain methods seem to struggle with the multimodal structure of the posterior. The advantage of using parallel tempering is clear. Table 1 compares three MCMC diagnostic statistics that indicate that the parallel-tempering sampler outperforms the single-chain algorithm.

Figure 2: **Top**: multiple trace plots for the redshift parameter, $z$. The uppermost graph is the trace of four independent parallel-tempering + slice-sampling (PT-SS) chains. Second graph depicts four independent slice-sampling (SS) only chains. These chains have clearly not converged (and are extremely slow to converge). **Bottom**: resulting histograms from the two methods (each trace is afforded its own color). It's clear that PT-SS is more consistently recovering posterior structure, whereas SS is getting stuck in small areas of the posterior. Note that the true redshift value is $z = 2.39$, located in the middle of the right mode of the well-sampled posterior.