[Reviews · NeurIPS 2015]

Submitted by Assigned_Reviewer_1

The paper introduces a model of a quasar's spectral energy distribution with GP priors on the raw basis functions. Two different data sources are combined: spectroscopic data from the BOSS survey and photometric data from the SDSS survey. Red-shift of the quasars is also a cornerstone of the model. Inference is done in two stages: the basis is fit using MAP and then MCMC with parallel tempering is used.

Quality The paper The paper proposes a model with quite a few design decisions e.g. K the number of basis functions, the fact that they be positive and normalised etc. It is not clear to which extent these choices are supported by empirical data or whether they are merely ad hoc. In the empirical evaluation, two aspects are missing: 1) There are no indications about runtime for training/fitting and testing neither in absolute terms neither compared to previous approaches. 2) It is not clear how much each of the two sources of information contributed to the predictive performance.

Clarity The paper is well written and well structured, the notation is good and the figures are meaningful.

Originality It seems that the model was designed almost from scratch bringing together various physical constraints. The two main competitors [2,3] seem to be based on multiple photometric datasets rather than on spectroscopy data. But as I don't know the relevant astronomical/cosmoligical literature, this is a just an educated guess. At least, I haven't seen a GP being a building block in a similar model.

Significance The increased data efficiency (as compared to previous approaches) is certainly a good feature of the method. As the paper deals with a particular application, the results are only relevant to a small fraction of the machine learning community.

Summary: The paper proposes a generative GP-based model for the spectral energy distribution of quasars aggregating spectral and photometric measurements. The paper seems to deal with a proper application of machine learning and the red-shift prediction experiments suggest a higher data efficiency of the method compared to previous approaches relying on photometry alone.

Submitted by Assigned_Reviewer_2

The paper presents a latent variable model for modeling spectral energy distribution of quasars, given stereoscopic and photometric observations. The joint modeling of stereoscopic and photometric measurements allows the model to make inferences about stereoscopic properties of quasars leveraging the more broadly available photometric data.

Clarity: The paper for the most part is well written and easy to follow. I have some minor complaints about the exposition, see detailed comments below.

Originality: Moderate. The authors develop a well motivated, non trivial latent variable model for capturing the salient properties of distributions of noisy quasar measurements. The use of parallel tempering in the inference procedure is interesting as well.

Significance: Although an application paper, the work is interesting and could spur further interest among the NIPS community, in tackling these important computational problems in astrophysics.

Detailed comments:

Model: 1) It would be useful to be explicit about the domains and the priors of the various random variables.

a) Is red shift variable z_n, constrained to take any real value in the domain [0, 7]. Is the prior on z_n, a uniform distribution in this interval?

b) What is the domain of the apparent brightness variable m_n, the positive real line? More importantly what prior was placed on m_n? Line 217 alludes to an empirically derived prior, but no further details are provided.

c) T he Dirichlet distribution is a standard prior on a collection of non negative variables that sum to one.

What is the more complicated mixture of log-Gaussian priors over w buying us here? A stronger motivation for this non standard, more complicated choice of prior will be helpful.

Inference: I found some of the details of the inference procedure confusing.

1) Basis estimation: It appears that the point estimate of the bases is being arrived at by finding a MAP estimate of an auxiliary model, one that ignores the photometric data. Why is the true model posterior which incorporates the photometric likelihood term p (y | w, z, m, B) in equation 10, not optimized instead? The MAP estimates of the two models are obviously different. When inferring w, z, and m, why does it make sense to condition on the MAP basis estimates of the auxiliary model? 2) It is interesting that the authors resort to parallel tempering. It is certainly plausible that a naive MCMC sampler will mix poorly. However, it would be good to clearly demonstrate the need for parallel tempering and show how much worse a single slice sampler chain (with T_c = 1) really is, in terms of red shift predictions. Perhaps an extra row could be added to Figure 5 and an additional line plot in Figure 6 to depict prediction results with and without parallel tempering.

Experiments: I like the two data splits with different train and test brightness and redshifts, mimicking real world scenarios.

The results could be better explained.

1) In figure (5) does the y-axis correspond to shifts inferred by the model? The rightmost scatter plot seems weird. 2) I found figure (6) very confusing.

It is never referenced in the text and it isn't clear what is being depicted in it. Are we looking at predicted SED (conditioning on photometric observations) for two held out quasars? Is the noisy line plot in light blue the observed SED for these quasars? What is the mysterious PCA-based model displayed in black, which seems to be fitting the data much better. Is it better because it is "cheating" and has been fit to all available SED's including the two SEDs displayed here?

Summary: Overall, I found the paper to be well written, interesting and above the publication threshold. However, there are some concerns that the authors need to address (see comments below).

Submitted by Assigned_Reviewer_3

This paper proposes a novel use of Gaussian process regression for astrophysical spectra estimation. The application to quasar spectra looks very interesting.

Due to the nature of spectral measurement in that domain, there are two different types of spectral data, which can be thought of as being generated by a common latent spectra. To model the generative process, they assume a couple of GPs sharing the latent spectra and some mixture weights.

Unlike the standard GP problem, the latent spectra is basically known up to the redshift parameter z. The main problem seems to be estimate the expectation of z as a function of optimized hyper-parameters, although the paper does not clearly explain so. The authors propose using a MAP estimation, whose relationship with the standard evidence approximation for hyper parameters are not very clear.

I found the model quite interesting. I strongly believe that this is a fantastic new application of machine learning (ML). The experimental results look interesting, too.

However, the main issue with this paper is that it does not fully explain the relationship with the existing model. Also, most of the descriptions are some sort of mixtures between physics and statistics/ML, making it very hard to follow. I'd suggest revising the paper to clearly explain - the relationship with existing models such as multitask learning - the problem setting itself - the relationship with the standard GPs (as I described above) - ...
Summary: Presents a very interesting application of Gaussian processes for spectra estimation in astrophysics. Methodological contributions are not clearly described, but seems to present a potentially interesting direction especially for applications where non-nevativity matters.

Author Feedback
Author rebuttal: Rev. 1 and 2:

- Using spectroscopic inferences (MAP basis/auxiliary model) in the photometric task reflects the astrophysical assumption that the SED of a quasar is highly structured, and leveraging information about this structure (from available spectroscopy) on a task with way less information (photometric redshift estimation) will improve estimates. The MAP component finds a useful representation of quasar spectra that the MCMC component can use for inference/predictions. Info from the MAP basis is critical.

Rev.1:

- Design decisions: we use nonnegative basis functions (BFs) to reflect the physical constraint of nonnegativity of spectral energies. We chose 4 BFs as an educated guess - 90% of variation can be described by 4 components in PCA (mentioned on page 3). We will validate K using test-reconstruction error in a final submission.
- Normalization of BFs is a choice of parameterization; the scale param m could be incorporated into the weights. We aimed to separate out overall scale from the SED itself, with the goal of picking up on structure invariant to overall brightness.
- We did not include runtime information - model output and implementations are quite different, MCMC can take a long time to converge so we ran many chains per test quasar with more samples than probably necessary. We also note that runtime, while of practical importance, is not a central issue in a scientific application like this where the goal is to accurately characterize the posterior to assess model performance. Approximately, XDSQSOz and NN took ~ hours each and our method took ~ days (but parallelized down to hours).

Rev.2:

- We will better describe parameter domains and priors in the final submission.
- cosmo redshift z > 0 (consistent with a monotonic expansion of the universe)
- A 'doppler effect' blueshift (z < 0) could occur for other objects
- z ~ N(3.5, 3^2) a priori (truncated at zero - aiming for uninformative but proper)
- log(m) ~ N(0, 10^2), m > 0. "empirically derived prior" chosen to be more dispersed than the sample of log(m) values derived from fitting the MAP basis.
- We chose MVLogisticNormal over Dirichlet for a few reasons. Representing weights as real-valued elicits an unconstrained optimization problem for the MAP basis. It was also easier to reason about, visualize, and fit structured priors on the pre-transform, real-valued variables. Also, this model doesn't gain anything from the nice properties of the Dirichlet (e.g. exp-family and conjugacy).
- Inference: you are correct - the true model posterior includes photometric information. We ignore photo data for basis inference based on the intuition that there is much more information about the basis in the spectro data than in the photo data. Optimizing over the integration term needed to compute the likelihood is a difficult problem, and sidestepped in this project. We will make this clearer in the exposition.
- Parallel Tempering (PT): for space, we did not write about the standard progression of simple MCMC (MH, etc) to complex MCMC (slice-sampling within PT). We will compare mixing diagnostics for naive MH vs PT.
- Experiments: we agree that this exposition can be improved, and will be on final submission.
- Figure 5: yes, the y-axis corresponds z inferred by the model; x-axis is a sort of ground truth (from spectro data). The rightmost scatterplot depicts results from the harshest data split - the basis trained here had zero quasars in test-range.
- Figure 6: Figure 6 is referenced on page 8 (line 401). You are correct - it is the predicted SED conditioned only on photometric observations (+ MAP basis). The noisy line plot is a spectroscopic measurement of the SED for that test quasar, and the PCA-based fit is a smoothed version of the spectro measurement. Indeed, it is "cheating" as it had access to the full spectro information for that quasar, whereas the red line is a single posterior sample given only photo data.

Rev.3:

- As an application paper, the model is designed for this unique situation, though it is similar to a smoothed nonnegative matrix factorization or factor analysis. We will expand on this comparison in the final draft.
- This model differs quite a bit from the "standard GP problem" - we learn a basis of positive functions (whose transformations have a GP prior) that describe a large set of functions.

Rev.6:

- Investigating model performance under train-test shifts is important because the sample of observable light sources is going to be biased toward brighter, closer objects. If the goal is to accurately characterize farther, older, fainter objects, then we want a model that is able to make predictions that lie outside of the range of the training sample.

General:

We would like to note that the intersection between astronomy and machine learning is a rich and fruitful area of research, and we hope to see a growing number of collaborations between the two communities!